# Why Do Donors Donate? A Study on Donation-Based Crowdfunding in Malaysia

Mohd Khairy Kamarudin [1,*], Nur Izzati Mohamad Norzilan [1], Fatin Nur Ainaa Mustaffa [1], Masyitah Khidzir [1], Suhaili Alma'amun [2], Nasrul Hisyam Nor Muhamad [1], Mohd Fauzi Abu-Hussin [1], Nurul Izzah Noor Zainan [3], Abdul Hafiz Abdullah [1] and Abdul Basit Samat-Darawi [1]

[1] Academy of Islamic Civilisation, Faculty of Social Sciences and Humanities, Universiti Teknologi Malaysia, Skudai 81310, Johor, Malaysia
[2] Faculty of Economics and Management, Universiti Kebangsaan Malaysia, Bangi 43600, Selangor, Malaysia
[3] Academy of Contemporary Islamic Studies, Universiti Teknologi MARA, Kelantan Branch 18500, Kelantan, Malaysia
* Correspondence: mohdkhairy@utm.my

**Abstract:** This study employed the Stimulus–Organism–Response (S-O-R) framework to investigate how social support and quality of the community affect the purpose to donate through donation-based crowdfunding. The online poll generated 359 responses, and the data were statistically analysed using the partial least square structural equation modelling (PLS-SEM) approach. Path coefficient analysis is also applied to figure out the outcomes of the relationships between the components. The results showed that service and system quality greatly influenced the donors' trust towards the donation-based crowdfunding. In addition, statistics showed that trust, quality of services, information value, and emotional support played a substantial role in explaining the donation purposes. The results could help donation-based crowdfunding platforms to enhance their success rate of donation campaigns. This study also provided a management application for each relationship and suggested helpful measures in attracting potential donors and retaining them.

**Keywords:** donation-based crowdfunding; social support; community quality; trust; donation purpose

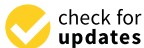



## 1. Introduction

A recent development shows that donation-based crowdfunding is the most frequent approach to pledge funds for charitable causes such as education projects, healthcare, natural disasters, and volunteerism. This approach collects donations from a large number of people through the Internet using diverse web-based platforms [1]. According to [2], online crowdfunding allows contributors to donate a small amount to help either individuals or small or large organisations. Researchers have argued that Internet and social media crowdfunding (Facebook, Instagram, WeChat, and Twitter) is an effective way to reach a big audience and gain broader support [3,4] and hence increase a project's success. This allows non-profit organisations to promote their humanitarian activities on social media via crowdfunding [5]. Generally, crowdfunding projects involve three parties: (i) the project initiator, who initiates a financing effort; (ii) supporters, who donate and share cash; and (iii) platforms, which link project initiators with contributors [6].

In Malaysia, donation-based crowdfunding websites were pioneered by Kitafund and Jomdonate. The websites enable anyone to create a cause and pledge for a specific amount of money to be collected within a given time period. Kitafund is a crowdfunding website to raise money for medical and humanitarian emergencies. As reported in April 2019, an estimated amount of RM 1.6 million was donated to people in need [7]. Jomdonate is a platform to channel assistance as a safeguard of welfare and human rights. In the

year 2022, JomDonate has a total of around RM 5 million with 220,000 contributors and 257 campaigns [8].

Despite the fact that web-based crowdfunding platforms offer great benefits and ease to raise funding for charitable causes, statistics show that, in Malaysia, there is a declining trend in the donation-based crowdfunding market. According to [9], Malaysia's donation-based crowdfunding business rose from USD 0.62 million in 2013 to USD 3.13 million in 2015. However, it plummeted to USD 1.68 million in 2016 and USD 0.04 million in 2017.

Due to crowdfunding's poor track record, a lack of study on the public's interest to donate raises some problems. Refs. [10–12] proposed that study on crowdfunding should concentrate on donors' intentions, motives, and behaviours, given that the potential donors are important to the success of a crowdfunding project. Recognising the factors that motivate donors to support crowdfunding projects is important to make use of new fundraising and networking opportunities [13]. Given this background, it is interesting to know the donors' motivations and answers to the following question, "What are the factors that influence the intention of donors to donate through crowdfunding platforms in Malaysia?"

In order to answer the research question, this study experimentally investigated the factors that influence the donors' intention. This study employed the Stimulus–Organism–Response (S-O-R) framework to identify crowdfunding's role in motivating the donors' behaviour, concentrating on the effect of community quality and social support on their intentions. The S-O-R framework is a behavioural study concept that is widely applied to interpret intentions [14]. This study also investigated the impacts of many organisms and behaviours (trust and donors' intention) in order to verify a new model. This approach introduced was the key driver behind this research, which was to explore the success of crowdfunding projects. It investigates the extended potential factor in the S-O-R framework for donation activities.

In gathering the data from the donors, this research employed a questionnaire survey instrument. The questionnaire was sent to the respective donors that were involved in crowdfunding projects. The data collected were then analysed descriptively and statistically using partial least square structural equation modelling (PLS-SEM) analysis. The PLS-SEM model permits estimation of a complex model with several concepts, variables, and structural approaches without statistical assumptions on the data. The PLS-SEM is a causal-predictive approach to SEM that emphasises prediction in estimating the statistical models whose structures are intended to offer explanations [15].

Hence, as for this study, it would be able to enhance the interest to donate in donation-based crowdfunding, and the breadth and content of the S-O-R framework are expected to be expanded. Furthermore, this study provided an empirical basis for broader research on online donors' intentions. This study also offered acceptable and persuasive fundraising campaign methods to boost project success rates in donation-based crowdfunding.

To further investigate the main research question, this paper analyses donation-based crowdfunding literature and the study model's theoretical foundation in Section 2; Section 3 follows, which presents the hypothesis and framework; Section 4 describes the study's methodology; Section 5 discusses the data analysis and findings; and finally, the last section provides the conclusion from this research and discusses the limitations as well as suggestions for future research.

## 2. Literature Review

### 2.1. Donation-Based Crowdfunding

According to the previous studies, there are several ways to identify the effective factors and goal of donation-based crowdfunding. Ref. [2] stated that personal networks and underlying project quality are associated with the success of crowdfunding efforts. One of the roles of donation-based crowdfunding platforms (CFP) is to support humanitarian and artistic projects [16], where the donors on donation-based CFPs can be viewed as philanthropists. The success degree of donation-based CFP depends on the quality

relationship between the "tastes" of the donors and the qualities of the project, as donations to a public benefit are the foundation of a donation-based campaign.

Furthermore, [17] believed that the role of trust in increasing donors' intention was downplayed. Donation-based crowdfunding platforms should include more social cues, such as photos, videos, and phrases, to enhance donors' trust. True words, photographs, and videos must also be used to urge contributors to check on the project's progression [18]. Other than trust, [18] demonstrated that social presence, attitude, perceived behavioural control, and personal norms are also connected with intention where similarly, [4] found a connection between attitude and perceived behavioural control. Otherwise, subjective standards have no bearing towards the donation intent. Donors are more likely to support crowdfunding campaigns when donors' desire to help is for themselves and the environment [13]. Ref. [18] explored extrinsic and intrinsic motivations in crowdfunding. In their study, social effect, generosity, and self-worth drove the crowdfunding contributions.

### 2.2. S-O-R Framework

The S-O-R framework is a meta-theory for studying user behaviour that has been shown in the fields of science and management information, with the aim of understanding customer loyalty, purchasing intention, behaviour, and engagement [14]. Based on this framework, which is based on environmental psychology, stimuli (S) increase people's cognitive and/or emotional reactions (O), which subsequently lead to behavioural responses (R) [19]. A person's underlying state (i.e., emotion) is assumed to be activated when they are subjected to stimuli. As a result, the person's intrinsic state influences their behavioural choices [20–22]. Ref. [10], enhanced the Stimulus–Organism–Response (S-O-R) framework by including empathy, perceived credibility, and relationship quality. Consequently, these three traits increase donors' intent.

In this framework, organisms respond to environment stimuli emotionally, prompting them to either approach or avoid [23,24]. A stimulus must be present so as to change an organism's mental and cognitive state or their internal responses to their surroundings [25,26]. According to [27], an organism's behaviour is dependent on how the environment affects it. "Organism" refers to a bodily reaction, condition, or feeling [28]. It is similar to the processing model, which focuses on how cognitive processes analyse environmental information that results in a choice [29]. It helps to improve a conceptual model to analyse and validate how individuals interact with an online platform (stimulus) that can engage and lead them (organism) with behavioural intention (response) [30,31].

The S-O-R framework is also recommended to evaluate the intention of online donation. Multiple sources of crowdfunding literature have proposed this framework to study project intention [32], funding behaviour [33], charitable crowdfunding intention [6,10,34,35], and fundraising success [36]. Therefore, in this study, the S-O-R framework is used to explain the effects of informational support, informational value, emotional support, service quality, system performance, and trust on crowdfunding donation. Organisms (beliefs) serve as a connection between stimuli (informational support, informational value, emotional support, service quality, and system performance) and the final outcome (donation intention). Based on this framework, human behavioural processes are also explained in order to forecast the online uses' cognitive judgements, behaviour, or intention.

## 3. Theoretical Framework and Hypothesis Development
### 3.1. Theoretical Framework

In this section, the theoretical framework is derived from the previous relevant literature. The theoretical framework for this study is then used to develop the hypotheses according to the variables suggested from the literature. This study proposed the following variables to be tested in this research, and these include social support, information quality, trust, and donation purposes. The following discussion formulates the hypothesis development based on the theoretical framework before a proposed research model is developed.

*3.2. Hypothesis Development*

3.2.1. Social Support's Impact on Trust and Intention to Donate

There are two types of social support: (1) a user perceived to be available to support and (2) a user that really supplies support [37]. Users who receive social assistance are more likely to develop a feeling of mutual duty, which leads them to assist other members of the community [38]. The concept of social support is based on intangible qualities such as informational and emotional elements [39].

Advice, feedback, suggestions, and recommendations from the community or peers that are useful in making any decision are referred to as informational support [40,41]. The informational support provides an exchange of information that affects online users' commitment [42–44]. According to [45] and [46], donors develop trust when informative support is meaningful and useful to helping the online community.

Hence, these findings supported the idea that donors who considered making a donation were more likely to look for the input and advice of other members among the crowdfunding community and to develop a sense of confidence in the platform. A number of recent studies [39,47,48] have shown that the availability of information may influence donors' purchase decisions or intentions. Therefore, this study argued that informational support has an impact on donors' willingness to donate. As a result, the following hypotheses are proposed:

**H1.** *Informational support is positively related to trust.*

**H2.** *Informational support is positively related to donation intention.*

Emotional support emphasises the expression of human compassion, which aids the resolution of problems in an indirect way [49]. Caring, trust, encouragement, recognition, and a feeling of community are all forms of emotional support [41]. They also influence decision-making behaviour [40], customer satisfaction [50], well-being [51], and social commerce intention [52,53]. As proposed by [53], emotional support affects trust. At the same time, [52] found that emotional support has a substantial association with social commerce intention. Thus, this study proposes hypotheses for trust in crowdfunding platforms which simultaneously have a major influence on donors' intention. The hypotheses are listed below.

**H3.** *Emotional support is positively related to trust.*

**H4.** *Emotional support is positively related to donation intention.*

3.2.2. Community Quality's Impact on Trust and Donation Intention

What makes a good community is determined by looking at how its users feel about its infrastructure. Ref. [46] recommended using information quality, system performance, and quality of service as the indicators of the community's quality. The proposed approach draws inspiration from the information systems success model created by [54].

Quality service means the quality of technical or online services offered to users [55,56]. The gap between what a client experiences and requires and what is given to meet those needs and expectations is measured by the quality of service [57]. High service quality consistently matches with users' expectation to foster trust among online users with e-services [58], financial information systems [59], online health communities [46], fintech [60], and mobile commerce [61]. In this case, a crowdfunding platform can be considered as an online system, with information and system quality being the key components in increasing the donors' intention to donate. Quality service also has a substantial impact on e-commerce [62], information systems [63], e-government [64], and mobile phone usage [65]. Hence, it can be concluded that the quality of service given by the crowdfunding platform is the foundation of donation-based crowdfunding. Below are the following hypotheses:

**H5.** *Quality service is positively related to trust.*

**H6.** *Quality service is positively related to donors' intention to donate.*

The quality of the platform's content is also one of the most essential characteristics [66]. Quality information is used to evaluate the success of online content as it is developed to facilitate its online commerce [67]. Recent studies have showed that the quality of information inspires trust in social commerce platforms [68,69], mobile commerce [61], e-commerce [62,70], and mobile banking [71]. As a result, this study considered that donors are more inclined to examine the quality of information when evaluating their trust in crowdfunding.

Meanwhile, information quality has directly impacted donor behaviour and intention in online platforms [63,72], fintech adoption [60], mobile systems [73,74], social media [75], digital libraries [76], and purchasing intention [56]. Hence, it is possible to hypothesise that information quality influences donation intention. The following hypotheses are shown below:

**H7.** *Quality of information is positively related to trust.*

**H8.** *Quality of information is positively related to donation intention.*

First impressions and the efficiency of a system have an influence on how the system is perceived by its user ([77]. The term "system quality" is used to describe the aforementioned qualities in an information system, as well as others such as usability, flexibility, and dependability [63]. Due to issues such as stagnant load times and unclear interfaces, donors may have a poor impression of the system's dependability [78]. System quality in the context of the platform defines the amount to which a system delivers on the features of its users' impressions. If the system quality is poor, the anticipated advantages are less likely to materialise [67]. As shown by previous research [1,60,61,77], the quality of the system increases the degree of trustworthiness among users. As a result of its demonstrated beneficial effect on online donors' satisfaction [55], continued usage [57,59,79], and intention [80,81], system quality is recognised as one of the reasons that contributes to behaviour [82]. Donors are swayed by the quality of a crowdfunding system because of the suggestions that were made in the prior study. Hypotheses were formed as below:

**H9.** *Quality of a system is positively related to trust.*

**H10.** *Quality of a system is positively related to donation intention.*

Trust or credibility, as an emotional and cognitive reaction, has the potential to influence people's judgments and behaviours. In order to increase donors' trust, platform design aspects (e.g., serviceability, navigation, and virtual/visual design) should be precise, comprehensive, and rationally arranged. It must be shown on the website [83]. It significantly influence purchase intentions [14,84,85], usage of mobile banking [86], and royalties in an online platform [87]. Ref. [49] discovered that donors' intention in medical crowdfunding is highly influenced by cognition-based and affect-based trust. Therefore, this study suggested trustworthiness as the central system of crowdfunding platforms as it was investigated through the view of potential donors' intentions to donate. The following hypothesis associated was proposed:

**H11.** *Trust is positively related to intention to donate.*

## 4. Research Model

According to the hypotheses generated, the research model is displayed in Figure 1. Based on the S-O-R framework, the model posited that donation intention (response) is favourably linked with trust (organism), social support, and community quality (stimulus), whereas the platform trust is strongly influenced by social and community support.

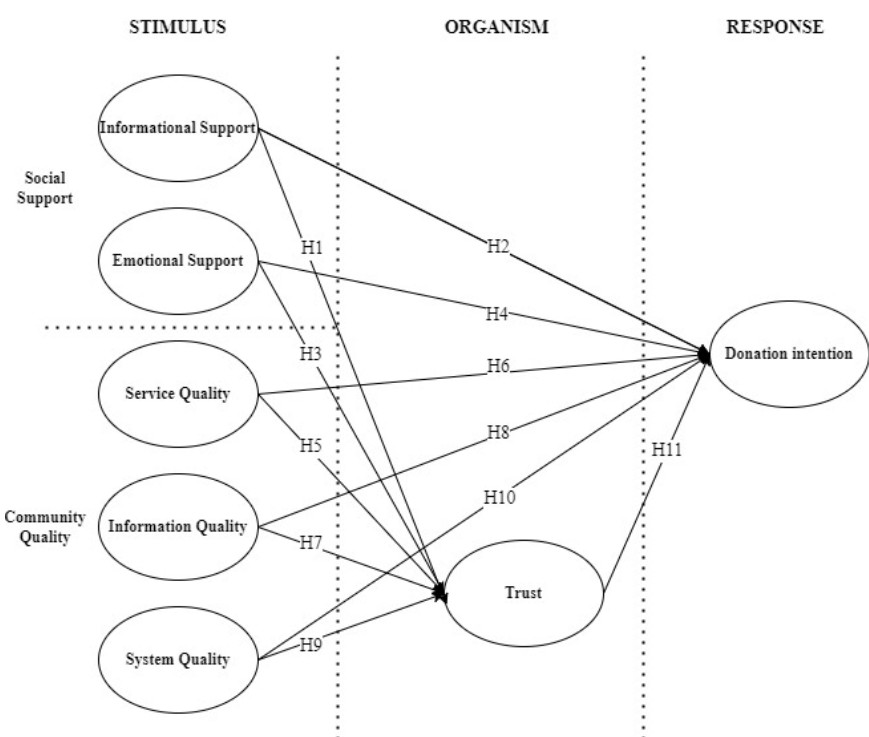

**Figure 1.** Research model. Source: modified from [46].

*4.1. Methodology*

4.1.1. Data Collection

All of the items in the questionnaire were used to measure the constructions that were developed from the previous study. It was designed with the perspectives of qualified scholars and related practitioners [30], and it was initially in English, translated into Malay, and then back-translated into English as practice by [23,24]. Two linguists assisted with the translation process to guarantee the content of the Malay version matches with the English version.

The questionnaire was separated into nine sections: the first section examined the respondents' demographic information (gender, age, educational background, and employment). The next eight sections investigated respondents' informational support, emotional support, information quality, quality of service, system performance, trust, and donors' intention. The items were obtained from various research and altered to fit with the study. A five-point Likert scale was used to assess the items (1: "Strongly Disagree" to 5: "Strongly Agree"). The item operationalization summary and the related references are presented in Appendix A Table A1.

The study took four months, from May 2022 to August 2022. The Google Form link was shared on the donation-based crowdfunding sites (Facebook and Instagram), and the respondents were based on the principle of voluntary participation as practiced by [14]. According to [22], this strategy saves money and time. A pilot study of 20 donors was undertaken prior to the distribution of the questionnaire. Its purpose was to verify the items in terms of reliability, meaning, and understandability [88]. A total of 359 respondents had taken part in the research by participating in answering the questionnaire. Table 1 exhibits the demographic information of the respondents.

As presented in Table 1, the majority of the respondents from this study were female participants (223, 62.1%). The age range was between 21 and 61, with the majority of the respondents being below 40 years old (78.6%). This tells us that the respondents come from the most active users of Internet services, and they are more familiar with the current use of online platforms. Our data also indicate that the majority of the respondents have at least a bachelor's degree and above, and only 83 (23.1%) hold a Malaysian Higher

School Certificate (STPM), which is equivalent to an A-Level certificate. This shows that the respondents from this research are so-called "educated" persons and working-class people (271; 75.5% of respondents are either with the government, private sector, or are self-employed). From the demographics perspective, this study suggests that the majority of the respondents are highly educated and employed, and only a few of them are students or unemployed with an education at only the school level.

**Table 1.** Demographics of the respondents ($N$ = 359).

| Demographic Profile | | Number ($N$ = 359) | Percentage (%) |
|---|---|---|---|
| **Gender** | Male | 136 | 37.9 |
| | Female | 223 | 62.1 |
| Age | 21–25 | 106 | 29.5 |
| | 26–30 | 75 | 20.9 |
| | 31–35 | 48 | 13.4 |
| | 36–40 | 53 | 14.8 |
| | 41–45 | 43 | 12.0 |
| | 46–50 | 18 | 5.0 |
| | 51–55 | 11 | 3.1 |
| | 56–60 | 4 | 1.1 |
| | 61 and above | 1 | 0.01 |
| Educational Level | 2 | 17 | 4.7 |
| | Malaysian Higher School Certificate (STPM) and equivalent | 83 | 23.1 |
| | Bachelor's degree and equivalent | 203 | 56.5 |
| | Master's degree and equivalent | 48 | 13.4 |
| | Ph.D. and equivalent | 8 | 2.2 |
| Marital Status | Single | 155 | 43.2 |
| | Married | 196 | 54.6 |
| | Divorced | 8 | 2.2 |
| Occupation | Government | 144 | 40.1 |
| | Private | 102 | 28.4 |
| | Self-employed | 25 | 7.0 |
| | Unemployed | 10 | 2.8 |
| | Student | 77 | 21.4 |
| | Retired | 1 | 3 |

### 4.1.2. Data Analysis

The data were analysed using Smart PLS 3.2.8. Smart PLS is a component-based structural equation modelling approach that handles both reflecting and formative structures. PLS-SEM provides a flexible and powerful approach for examining relationships between latent variables and indicators. PLS-SEM was chosen as the main objective of this paper to anticipate the crucial constructs and to handle a complicated research model [15].

### 4.1.3. Model Measurement

The test's reliability displayed the measuring scale's dependability, consistency, and stability, which were often verified using Cronbach's alpha and composite reliability (CR). Cronbach's alpha and CR must be more than 0.70 to be considered appropriate [89]. This study found that Cronbach's alpha and CR values for each construct were more than 0.7, which implies that each item was linked to its associated construct. Since the measurement components in this model were generated from relevant previous studies and validated by the experts, the measuring scale is deemed to have sufficient content validity.

The extracted average variance (AVE) for all constructs should be more than 0.50 to be deemed as acceptable [15,89]. The AVE of each item in this study was varied from 0.725 to 0.860, with factor loadings ranging from 0.798 to 0.940. This showed that a satisfactory

degree of convergent validity was achieved. Table 2 presents the reliability and convergent validity of the items.

**Table 2.** Reliability and convergent validity.

| Construct | Measurement | Mean | Factor Loadings | Cronbach's $\alpha$ | CR | AVE |
|-----------|-------------|------|-----------------|---------------------|-----|-----|
| Informational Support | InfoS1 | 3.83 | 0.915 | 0.919 | 0.949 | 0.860 |
| | InfoS2 | 3.84 | 0.940 | | | |
| | InfoS3 | 3.84 | 0.928 | | | |
| Emotional Support | EmoS1 | 3.91 | 0.921 | 0.889 | 0.931 | 0.818 |
| | EmoS2 | 3.94 | 0.917 | | | |
| | EmoS3 | 3.69 | 0.876 | | | |
| Service Quality | ServQ1 | 3.87 | 0.849 | 0.873 | 0.913 | 0.725 |
| | ServQ2 | 3.79 | 0.855 | | | |
| | ServQ3 | 3.86 | 0.900 | | | |
| | ServQ4 | 3.68 | 0.798 | | | |
| Information Quality | InfoQ1 | 3.96 | 0.864 | 0.918 | 0.942 | 0.803 |
| | InfoQ2 | 3.87 | 0.906 | | | |
| | InfoQ3 | 3.88 | 0.902 | | | |
| | InfoQ4 | 3.92 | 0.913 | | | |
| System Quality | SysQ1 | 3.89 | 0.875 | 0.914 | 0.939 | 0.794 |
| | SysQ2 | 3.99 | 0.887 | | | |
| | SysQ3 | 4.02 | 0.917 | | | |
| | SysQ4 | 3.91 | 0.886 | | | |
| Trust | Trust1 | 3.90 | 0.825 | 0.944 | 0.955 | 0.781 |
| | Trust2 | 4.01 | 0.902 | | | |
| | Trust3 | 3.89 | 0.895 | | | |
| | Trust4 | 3.96 | 0.908 | | | |
| | Trust5 | 3.87 | 0.906 | | | |
| | Trust | 3.96 | 0.863 | | | |
| Donation Intention | Intent1 | 4.04 | 0.891 | 0.927 | 0.945 | 0.774 |
| | Intent2 | 3.84 | 0.855 | | | |
| | Intent3 | 4.09 | 0.890 | | | |
| | Intent4 | 4.13 | 0.884 | | | |
| | Intent5 | 4.14 | 0.879 | | | |

Note(s): at the 0.001 level, factor loadings are significant.

Furthermore, the correlations between each reflective concept and the constructs are much less than the square root of the AVE, demonstrating the discrimination validity. The heterotrait–monotrait ratio of correlations (HTMT) criteria were also used to evaluate discriminant validity, with HTMT confidence intervals greater than 0.90 [89]. The findings demonstrated that none of the HTMT confidence intervals were more than 0.90, showing that there is no discrimination between the items. Table 3 demonstrates the test result of discriminant validity.

**Table 3.** Test results of discriminant validity.

| | Informational Support | Emotional Support | Service Quality | Information Quality | System Quality | Trust | Donation Intention |
|---|---|---|---|---|---|---|---|
| Informational Support | | | | | | | |
| Emotional Support | 0.789 | | | | | | |
| Service Quality | 0.669 | 0.684 | | | | | |
| Information Quality | 0.676 | 0.715 | 0.855 | | | | |
| System Quality | 0.681 | 0.700 | 0.721 | 0.824 | | | |
| Trust | 0.556 | 0.563 | 0.672 | 0.638 | 0.597 | | |
| Donation Intention | 0.586 | 0.637 | 0.683 | 0.672 | 0.605 | 0.760 | |

#### 4.1.4. Structural Model

This research determined the validity and reliability before evaluating the structural model. The collinearity test was used to ensure that there was no issue with multicollinearity, as indicated in Table 4. There was no indication of multicollinearity in this study since the VIF values were less than 5 [89]. Other than that, this study used the bootstrapping approach to assess the relevance of the path coefficient in order to evaluate the outcomes of the structural model.

**Table 4.** Test results of all hypotheses.

| Hypotheses | Path | Path Coefficient | t-Statistic | *p*-Value | Conclusion | VIF |
|---|---|---|---|---|---|---|
| **H1** | Informational Support → Trust | 0.104 | 1.504 | 0.066 | Not supported | 2.371 |
| **H2** | Informational Support → Donation Intention | 0.034 | 0.557 | 0.289 | Not supported | 2.390 |
| **H3** | Emotional Support → Trust | 0.080 | 1.146 | 0.126 | Not supported | 2.445 |
| **H4** | Emotional Support → Donation Intention | 0.159 | 2.575 | 0.005 | Supported | 2.456 |
| **H5** | Service Quality → Trust | 0.305 | 3.993 | 0.000 | Supported | 2.647 |
| **H6** | Service Quality → Donation Intention | 0.117 | 1.820 | 0.034 | Supported | 2.813 |
| **H7** | Information Quality → Trust | 0.145 | 1.596 | 0.056 | Not supported | 3.499 |
| **H8** | Information Quality → Donation Intention | 0.122 | 1.792 | 0.037 | Supported | 3.537 |
| **H9** | System Quality → Trust | 0.134 | 1.784 | 0.037 | Supported | 2.625 |
| **H10** | System Quality → Donation Intention | 0.012 | 0.191 | 0.424 | Not supported | 2.657 |
| **H11** | Trust → Donation Intention | 0.461 | 8.070 | 0.000 | Supported | 1.796 |

### 5. Result

This study presented four hypotheses for social support (H1, H2, H3, and H4). However, only one hypothesis, H4, is statistically significant. This demonstrates that emotional support has a favourable effect on donor intention through donation-based crowdfunding platforms ($\beta = 0.159$, $p < 0.05$). Furthermore, informational support does not correlate with trust or donor intention, as emotional support is not correlated with trust.

Table 4 shows that the structural model's findings support four out of the six hypotheses for community quality. Quality service is advocated for by having a favourable influence on its credibility ($\beta = 0.305$, $p < 0.001$) and donation intention ($\beta = 0.117$, $p < 0.05$). Then, information quality that is related to donor intention ($\beta = 0.122$, $p < 0.05$) is not related to trust in crowdfunding. Figure 2 demonstrates that system quality has a substantial influence on trust ($\beta = 0.134$, $p < 0.05$) but not on donor intention. As proposed by H11, trust has a favourable effect on donor intention through the platform ($\beta = 0.461$, $p < 0.001$).

Figure 2 presents how much of the changes in the endogenous variables can be explained by factors outside of the system. It is used to assess the overall predictive capability of the model. Through donation crowdfunding, the model explained about

44.3% percent of the variation in trust and 59.5% percent of the variance in donation intention.

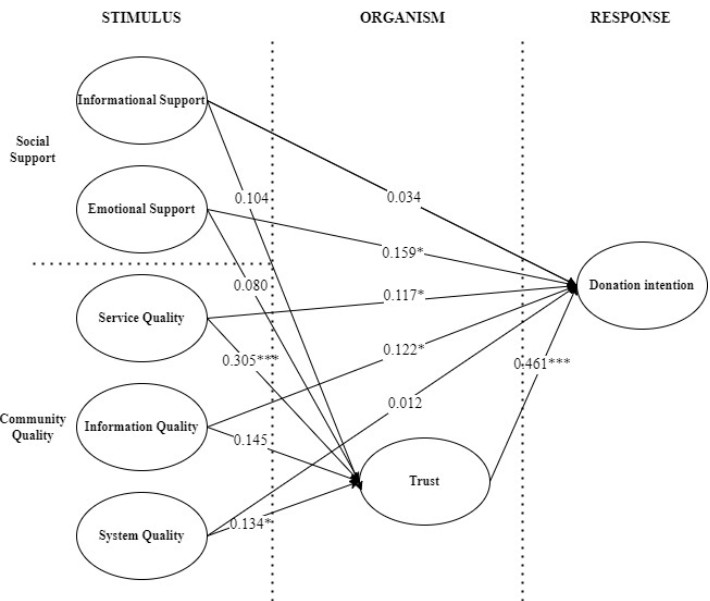

**Figure 2.** The research model's test findings. **Note(s):** *: $p < 0.05$; ***: $p < 0.001$.

## 6. Discussion and Implications

PLS-SEM research shows that crowdfunding credibility is linked with service and system quality. Instead of motivation [34] and credibility [10,35], emotional support, quality service, information quality, and trust also influence potential donors' intentions. They are more inclined to donate if they receive emotional support, quality service, reliable information, and trust. The results attained can boost the success of donation-based crowdfunding campaigns with the application of the S-O-R framework. This statement suggests that the respondents' cultural background has an impact on their perception of trust and emotional support in crowdfunding. The popularity of crowdfunding among Muslims, through the promotion of *Sadaqah*, Gift, and Zakat, highlights the importance of these two variables.

Furthermore, the study's results help funders and service providers by explaining the factors that influence donor's intentions. Ref. [50] found that community emotional support is important. Community support increases current and potential donors' intentions [48], whereas social support helps individuals feel better and stay optimistic. Emotional support from others is needed when donors have a negative view of crowdsourcing. They will use social media to collect information, relying more on peer users than corporate content [39]. Contrary to the initial study's predictions, informational support did not alter their purpose.

Given that service quality is linked with trust and intention, donation-based crowdfunding sites will focus on improving their services and campaigns [90]. Keeping the website updated, well-structured posting, easy-to-navigate content, maintaining online transaction and personal information security, and marketing new services are several other ways to improve the service quality [58]. Crowdfunding platforms must build donors' confidence in online donation security so they can transact safely through their devices [91].

According to past studies, information quality is correlated with intention [14,77,92]. The crowdfunding platforms should provide relevant, adequate, precise, and up-to-date information about their goods and services. Other studies also demonstrated that information quality does not affect crowdfunding trust [59,61,66]. Ref. [93] suggested donors will trust an organisation if it outlines its full operation.

Moreover, the quality of the platform enhances its credibility but not a person's desire to donate [57,60,94]. The system's accessibility and interactive interface can boost donors'

confidence. The organisation must demonstrate user-friendly systems with well-designed navigation and aesthetic appeal to boost its quality [92]. The system's professionalism and dependability also will enhance donors' confidence [12]. Ref. [59] established a relationship between system quality and planned behaviour. However, this study differs.

Finally, trust also motivates donations [86,95]. Donors usually evaluate a platform's campaign to ensure their money will not misused [10]. This study proposes that an easy-to-use donation system will increase platform confidence. Along with consistent communication with donors, high social presence can also strengthen donors' trust and their perceptions of its legitimacy [10]. The CFP providers can consider the following suggestions for improving their systems and making them more user friendly:

1. Mobile-First Platforms: a user-friendly crowdfunding system should have a mobile-optimized platform, allowing for seamless use and accessibility from any device.
2. Automated Processes: automated processes such as automatic distribution of funds, automatic email notifications, and automatic project updates.
3. Personalized Dashboards: a personalized dashboard for organizers and contributors that displays real-time information about the project and its progress, such as the amount raised and the status of causes.

## 7. Limitations and Future Research Suggestion

This research has come to the conclusion that several factors influence donors' motivation to contribute their money to the web-based crowdfunding platforms in Malaysia. Given the factors that were found in this research, as discussed in the previous section, this paper admits that there are several limitations that need to be considered. As this research is limited to respondents in Malaysia, conclusions and findings from this research are not to be generalized. On top of that, the respondents may not represent a typical structure of donors in Malaysia or elsewhere. The over-representation of young adults with tertiary education and government employees in the sample may limit the external validity of the findings, as the results may not be generalizable to other groups of donors. This bias can impact the external validity of the findings, meaning that the results cannot be applied to the general population of donors or to other countries. To increase the external validity, a more diverse sample of donors should be included in future research. On the other hand, for future research, the focus could be on examining rewards-based, debt-based, and international crowdfunding platforms. On top of that, it would be interesting to investigate the influence of cultural variations in determining the intention to donate. The study could also be expanded by looking at donation's intentions in other cultural contexts as well as analysing how social support and a community's standards influence the donors' trust and intentions. It would be interesting to study other stimuli and organism characteristics as well, such as the platform's attractiveness, visibility, donors' experiences and contentment, religion, and attitude.

**Author Contributions:** Conceptualization, M.F.A.-H.; Methodology, M.K.K.; Validation, A.H.A.; Formal analysis, F.N.A.M.; Investigation, M.K. and A.B.S.-D.; Resources, N.I.M.N.; Writing—original draft, N.I.N.Z.; Writing—review & editing, N.H.N.M.; Funding acquisition, S.A. All authors have read and agreed to the published version of the manuscript.

**Funding:** This work was funded by the Universiti Teknologi Malaysia under UTM Encouragement Research (PY/2020/04097).

**Institutional Review Board Statement:** The authors adhered to the ethical guidelines of the Committee on Publication Ethics (COPE).

**Informed Consent Statement:** Not applicable.

**Data Availability Statement:** Data available in a publicly accessible repository.

**Conflicts of Interest:** There are no conflict of interest to declare.

# Appendix A

**Table A1.** Research items.

| Construct | ID | Measure | Adapted from |
|---|---|---|---|
| Informational Support | InfoS1 | When I wish to donate, some people give me the information about crowdfunding. | [46,50,96] |
| | InfoS2 | When I wish to donate, some people will suggest donation-based crowdfunding platform. | |
| | InfoS3 | When I wish to donate, some people will help me to identify potential donation-based crowdfunding platform. | |
| Emotional Support | EmoS1 | When I wish to donate in crowdfunding, some people around me support me. | [46,50,96] |
| | EmoS2 | When I wish to donate in crowdfunding, people around me encourage me. | |
| | EmoS3 | When I wish to donate in crowdfunding, people around me will concern me. | |
| Service Quality | ServQ1 | Crowdfunding platform provides on-time services. | [46,92,97] |
| | ServQ2 | Crowdfunding platform provides prompt responses. | |
| | ServQ3 | Crowdfunding platform provides professional services. | |
| | ServQ4 | Crowdfunding platform provides personalized services. | |
| Information Quality | InfoQ1 | Crowdfunding website provides relevant information. | [46,92,97] |
| | InfoQ2 | Crowdfunding website provides sufficient information. | |
| | InfoQ3 | Crowdfunding website provides accurate information. | |
| | InfoQ4 | Crowdfunding website provides up-to-date information. | |
| System Quality | SysQ1 | I feel that crowdfunding website exhibits nice graphics. | [46,92,97] |
| | SysQ2 | I feel that crowdfunding website is easy to use. | |
| | SysQ3 | I feel that crowdfunding website is easy to navigate. | |
| | SysQ4 | I think crowdfunding website is visually attractive. | |
| Trust | Trust1 | I trust crowdfunding platform to do what they promise. | [68,84] |
| | Trust2 | I trust initiator/project creator to do what they promise. | |
| | Trust3 | I believe crowdfunding platforms are reliable. | |
| | Trust4 | I believe crowdfunding platforms are dependable. | |
| | Trust5 | I believe crowdfunding platforms are genuinely committed to my satisfaction. | |
| | Trust6 | Overall, I can trust crowdfunding platforms for making a donation. | |
| Donation Intention | Intent1 | Given the chance, I intend to donate in crowdfunding. | [12,34] |
| | Intent2 | I intend to actively donate in crowdfunding. | |
| | Intent3 | I expect to donate in crowdfunding in the future. | |
| | Intent4 | I would use the donation-based crowdfunding platform to help others. | |
| | Intent5 | I am willing to make donations to good projects on the platform. | |

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
