# Peer review of "Why Do Donors Donate? A Study on Donation-Based Crowdfunding in Malaysia"

_sustainability, doi:10.3390/su15054301_

Round 1
Reviewer 1 Report
I appreciate that the paper is interesting, deals with an important issue and offers plausible results. However, the paper has some aspects that can be improved. The main drawback of the article are the following:
1. The application of path analysis is mentioned in the abstract, but within the study, the results of SEM modelling are presented.
2. The Introduction section should emphasize more clearly the element of novelty that the study brings in the context of specialized literature.
3. It is necessary to justify the choice of the factors "social support" and "quality of community" in relation to "donation-based crowdfunding" to the detriment of other factors.
4. Some abbreviations are used in the paper without indicating what they refer to (e.g., CFP, HTMT, SPM)
5. In the section "3. Hypothesis Development and Theoretical Framework" only subsection "3.1. Hypothesis Development" is defined. I recommend the addition of the subsection "3.2. Theoretical Framework" in which the elements from lines 270-276 should be included.
6. In Figure 1, it is necessary to indicate the source.
7. In section 4, the description of some important elements related to data and methods is missing. In subsection "4.1.1. Data collection" it is mentioned that the demographic characteristics of the respondents are presented in Table 2. It is necessary to describe the results included in the table. Also, in subsection "4.1.2. Data analysis", the presentation of the methods applied in the study (e.g., PL-SEM, path analysis) is important.
8. Line 268: should be "H11" instead of "H9"
9. Line 329: should be "Table 5" instead of "Table 8"
10. In addition to describing the results of testing the significance of the coefficients, it is also necessary to present the results related to Goodness-of-Fit indices.
11. The Conclusions section should be included.
Good luck!
Author Response
No |
Comment |
Notes |
|
1. The application of path analysis is mentioned in the abstract, but within the study, the results of SEM modelling are presented. |
The path co-efficient analysis is presented with the SEM analysis result. These are discussed in page 362 from row 361. The abstract was also modified. |
|
2. The Introduction section should emphasize more clearly the element of novelty that the study brings in the context of specialized literature. |
Sentences in Line 72 – 74 were added in the article. |
|
3. It is necessary to justify the choice of the factors "social support" and "quality of community" in relation to "donation-based crowdfunding" to the detriment of other factors. |
Based on LR that we discussed in the introduction and hypothesis development, , we found that these two factors were the potential factors of the subject studied. This is discussed in line 162 – 164 |
|
4. Some abbreviations are used in the paper without indicating what they refer to (e.g., CFP, HTMT, SPM) |
CFP - Crowdfunding Platforms HTMT - Heterotrait-monotrait ratio of correlations SPM - Malaysian Certificate of Education STPM - Malaysian Higher School Certificate |
|
5. In the section "3. Hypothesis Development and Theoretical Framework" only subsection "3.1. Hypothesis Development" is defined. I recommend the addition of the subsection "3.2. Theoretical Framework" in which the elements from lines 270-276 should be included. |
Thanks for the suggestion. We start with theoretical framework in 3.1 instead before discuss the Hypothesis Development in 3.2. |
|
6. In Figure 1, it is necessary to indicate the source. |
|
|
7. In section 4, the description of some important elements related to data and methods is missing. In subsection "4.1.1. Data collection" it is mentioned that the demographic characteristics of the respondents are presented in Table 2. It is necessary to describe the results included in the table. Also, in subsection "4.1.2. Data analysis", the presentation of the methods applied in the study (e.g., PL-SEM, path analysis) is important. |
This has been modified and added. |
|
8. Line 268: should be "H11" instead of "H9" |
corrected |
|
9. Line 329: should be "Table 5" instead of "Table 8" |
corrected |
|
10. In addition to describing the results of testing the significance of the coefficients, it is also necessary to present the results related to Goodness-of-Fit indices. |
This study did not present Goodness of fit indices as proposed by (Hair et al., 2019, pp. 572) Hair, J. F., Sarstedt, M., & Ringle, C. M. (2019). Rethinking some of the rethinking of partial least squares. European Journal of Marketing, 53(4), 566–584. https://doi.org/10.1108/EJM-10-2018-0665
|
Reviewer 2 Report
Article reviewed regards contemporary issue in social activity which is crowdfunding (in case of Malaysia). It's topic expressing important aspect of financing in today's society and business. This activity is realized by specific technological tools creating some kind of environment together with participants and their behaviors.
Authors formulate research problem and discuss it in the light of theoretical background. They cite wide set of significant and current literature (over 100 items). The same situation is related to study design - it's based on proposal presented in literature and used for paper goals. Its design, procedure, methods used and results are discussed in proper way as well. Readers can understand authors way of thinking, their aims in this study and final decision taken on hypotheses verification. Hypotheses are formulated on current state of research achievements presented in literature too.
Authors declare limitations and future research suggestions. These ideas expresses important issues. It's worth mentioning that Authors raise problem of socio-cultural context for study results and future research activities. That's why it is important to raise the issue of cultural context in discussion and implications section. My suggestion for improvement of the paper is to incorporate this issue in case of Malaysian society in discussion on results achieved (for example is there any cultural indications which may explain hypotheses positive or negative verification in opinions of the Authors?). In present form this section of the paper focusing only on the state of worldwide literature.
Author Response
No |
Comment |
Notes |
|
Article reviewed regards contemporary issue in social activity which is crowdfunding (in case of Malaysia). It's topic expressing important aspect of financing in today's society and business. This activity is realized by specific technological tools creating some kind of environment together with participants and their behaviors. |
Thanks for the compliment |
|
Authors formulate research problem and discuss it in the light of theoretical background. They cite wide set of significant and current literature (over 100 items). The same situation is related to study design - it's based on proposal presented in literature and used for paper goals. Its design, procedure, methods used and results are discussed in proper way as well. Readers can understand authors way of thinking, their aims in this study and final decision taken on hypotheses verification. Hypotheses are formulated on current state of research achievements presented in literature too. |
Thanks for the compliment |
|
Authors declare limitations and future research suggestions. These ideas expresses important issues. It's worth mentioning that Authors raise problem of socio-cultural context for study results and future research activities. That's why it is important to raise the issue of cultural context in discussion and implications section. My suggestion for improvement of the paper is to incorporate this issue in case of Malaysian society in discussion on results achieved (for example is there any cultural indications which may explain hypotheses positive or negative verification in opinions of the Authors?). In present form this section of the paper focusing only on the state of worldwide literature. |
This is added in line 396 - 399 |
Reviewer 3 Report
The study on donation-based crowdfunding applies the standard partial least square structural equation modelling (PLS-SEM) method to establish relation between the service and system quality, trust, and informational and emotional support on one hand and donation-based crowdfunding on the other hand. An online poll (N = 359) is key source of data.
Hypotheses are formulated in a clear way. The PLS-SEM model is well-executed. The measurement and structural models are well described.
Key comments relate to data structure and data limitations.
· Donation is subject to self-selection of donors. Table 2 (p. 8), for example, suggest that people in age group 21-30 accounted for half of the sample. Three quarters of the survey participants reported tertiary education and over 40 percent were government employees. Is it a typical structure of donors (in general donations and in crowdfunding in particular) in Malysia and elsewhere? If not, how the self-selection impacts external validity of their findings?
· The section on Limitations (p. 12) only states that ‘this research is limited to the respondents in Malaysia, conclusions and findings from this research are not to be generalized’. But it is unclear, whether the study is representative for the Malaysia as well.
· Finally, the section on ‘Discussion and Implications’ is rather vague about practical recommendation for the crowdfunding. It, for example, concluded (p. 12) that ‘an easy-to-use donation system will increase platform confidence’. The authors are encouraged to provide some more specific recommendation for more efficient and user-friendly crowdfunding systems.
Author Response
No |
Comment |
Notes |
|
The study on donation-based crowdfunding applies the standard partial least square structural equation modelling (PLS-SEM) method to establish relation between the service and system quality, trust, and informational and emotional support on one hand and donation-based crowdfunding on the other hand. An online poll (N = 359) is key source of data. Hypotheses are formulated in a clear way. The PLS-SEM model is well-executed. The measurement and structural models are well described. Key comments relate to data structure and data limitations. |
Thanks for the compliment |
|
Donation is subject to self-selection of donors. Table 2 (p. 8), for example, suggest that people in age group 21-30 accounted for half of the sample. Three quarters of the survey participants reported tertiary education and over 40 percent were government employees. Is it a typical structure of donors (in general donations and in crowdfunding in particular) in Malysia and elsewhere? If not, how the self-selection impacts external validity of their findings? |
Discussion on this is further elaborate in limitations. |
|
· The section on Limitations (p. 12) only states that ‘this research is limited to the respondents in Malaysia, conclusions and findings from this research are not to be generalized’. But it is unclear, whether the study is representative for the Malaysia as well.
|
Given the limitation of the numbers of respondents which was distributed in Malaysia, this research can only be say applicable to Malaysian context. However, this research can be replicated to another context be it other culture, population and etc. |
|
Finally, the section on ‘Discussion and Implications’ is rather vague about practical recommendation for the crowdfunding. It, for example, concluded (p. 12) that ‘an easy-to-use donation system will increase platform confidence’. The authors are encouraged to provide some more specific recommendation for more efficient and user-friendly crowdfunding systems.
|
The suggestion is accepted. Correction has been done in page 12. |
Round 2
Reviewer 1 Report
I think the authors successfully managed to address the reviewer's recommendations. I appreciate that the current version can be published. Good luck with your future research!
Reviewer 3 Report
Paper was improved. Most comments were processed.